# DNA-PK promotes DNA end resection at DNA double strand breaks in $G_0$ cells

**Faith C Fowler[1,2], Bo-Ruei Chen[3], Nicholas Zolnerowich[4], Wei Wu[4], Raphael Pavani[4], Jacob Paiano[4], Chelsea Peart[2], Zulong Chen[2], André Nussenzweig[4], Barry P Sleckman[3]\*, Jessica K Tyler[2]\***

[1]Weill Cornell Medicine Pharmacology Graduate Program, New York, United States; [2]Weill Cornell Medicine, Department of Pathology and Laboratory Medicine, New York, United States; [3]Department of Medicine, Division of Hematology and Oncology, O'Neal Comprehensive Cancer Center, University of Alabama at Birmingham, Birmingham, United States; [4]Laboratory of Genome Integrity, National Cancer Institute, Bethesda, United States

**Abstract** DNA double-strand break (DSB) repair by homologous recombination is confined to the S and $G_2$ phases of the cell cycle partly due to 53BP1 antagonizing DNA end resection in $G_1$ phase and non-cycling quiescent ($G_0$) cells where DSBs are predominately repaired by non-homologous end joining (NHEJ). Unexpectedly, we uncovered extensive MRE11- and CtIP-dependent DNA end resection at DSBs in $G_0$ murine and human cells. A whole genome CRISPR/Cas9 screen revealed the DNA-dependent kinase (DNA-PK) complex as a key factor in promoting DNA end resection in $G_0$ cells. In agreement, depletion of FBXL12, which promotes ubiquitylation and removal of the KU70/KU80 subunits of DNA-PK from DSBs, promotes even more extensive resection in $G_0$ cells. In contrast, a requirement for DNA-PK in promoting DNA end resection in proliferating cells at the $G_1$ or $G_2$ phase of the cell cycle was not observed. Our findings establish that DNA-PK uniquely promotes DNA end resection in $G_0$, but not in $G_1$ or $G_2$ phase cells, which has important implications for DNA DSB repair in quiescent cells.

**\*For correspondence:**
bps@uab.edu (BPS);
jet2021@med.cornell.edu (JKT)

## Editor's evaluation

This manuscript will be of relevance to scientists interested in cell cycle, DNA repair, and genome stability reporting the unexpected discovery that the DNA-dependent protein kinase (DNA-PK) is required for DSB resection in G0 cells, whereas it is known and confirmed here that it inhibits resection in G1 and G2 cells. This finding has important implications for the clinical application of DNA-PK-targeted inhibitors. The data are of high quality and derive from two independent cell lines, genetic requirements were mostly established by gene knockouts, and the latest genome-wide sequencing techniques were applied to measure resection tracts. The key claims of the manuscript are supported by the data presented by the authors.

## Introduction

DNA double-strand breaks (DSBs) are particularly deleterious lesions which, if left unrepaired, can lead to cell death, or if repaired aberrantly, can lead to oncogenic chromosomal translocations and deletions (*Jackson and Bartek, 2009*). Eukaryotic cells utilize two main mechanisms of DSB repair: non-homologous end joining (NHEJ), where the broken DNA ends are ligated together with minimal processing of the DNA termini; and homologous recombination (HR), which uses a homologous sequence, usually on a sister chromatid, as a template for accurate DNA repair. Because HR relies on

a homologous template for accurate repair, HR is mostly restricted to S and $G_2$ phases of the cell cycle when sister chromatids exist. On the other hand, cells can employ NHEJ in any phase of the cell cycle and it is the only option in quiescent ($G_0$) cells and $G_1$ phase cells (*Scully et al., 2019*).

Extensive DNA end resection of the broken DNA ends, which generates long tracts of 3′ ssDNA overhangs at DSBs, is a critical step in committing the cell to use HR to repair DSBs. DNA end resection is initiated by nucleases MRE11 and CtIP, and subsequently extended by nucleases including EXO1 and DNA2/BLM (*Paull and Gellert, 1998*; *Trujillo et al., 1998*; *Sartori et al., 2007*; *Gravel et al., 2008*; *Mimitou and Symington, 2008*; *Zhu et al., 2008*; *Bunting et al., 2010*). The 3′ ssDNA overhangs are quickly bound by the single-stranded binding protein trimer replication protein A (RPA) to stabilize and protect the ssDNA, and later in repair RPA is replaced by the RAD51 recombinase protein that leads to the homology search to find a homologous template to achieve accurate HR repair (*Sugiyama and Kowalczykowski, 2002*; *San Filippo et al., 2008*; *Wright et al., 2018*). NHEJ is initiated by the KU70/KU80 heterodimer binding to broken DNA ends (*Zahid et al., 2021*). KU70/KU80 recruits the DNA-dependent protein kinase catalytic subunit (DNA-PKcs) which together form a complex called DNA-PK (*Gottlieb and Jackson, 1993*; *Hammarsten and Chu, 1998*). Once the DNA-PK complex is formed, the KU heterodimer translocates inwards along the DNA and DNA-PKcs remains at the DNA ends, undergoing activation via conformational changes mediated by autophosphorylation of the ABCDE cluster (*Yaneva et al., 1997*; *Chen et al., 2021b*). Recent cryo-EM structures of DNA-PK also implicate dimerization of DNA-PK as important in recruiting downstream NHEJ factors by bringing broken DNA ends together (*Chaplin et al., 2021*; *Zha et al., 2021*). In addition to autophosphorylation, DNA-PKcs phosphorylates members of the NHEJ machinery, including the KU heterodimer, XRCC4, XLF, and Artemis (*Bartlett and Lees-Miller, 2018*).

The critical bifurcation point in the choice to use HR or NHEJ to repair DSBs is the processing of broken DNA ends to form single-stranded 3′ DNA overhangs, which blocks NHEJ and commits the cell to HR (*Symington and Gautier, 2011*). Therefore, DNA end resection is tightly regulated to prevent aberrant DNA end resection in $G_0$ and $G_1$ phase cells, where NHEJ is the major DSB repair pathway. Several factors have been identified as critical DNA end protection factors that limit resection of DNA DSBs including 53BP1, RIF1, and the Shieldin complex. The proposed mechanism of action of 53BP1 and its downstream effectors include acting as a physical barrier to protect DNA ends from nucleases and promoting DNA polymerase α activity to quickly fill in any resected ends (*Bunting et al., 2010*; *Dev et al., 2018*; *Mirman et al., 2018*; *Noordermeer et al., 2018*; *Setiaputra and Durocher, 2019*; *Paiano et al., 2021*). Additionally, KU70/KU80 has also been shown in budding yeast *Saccharomyces cerevisiae* to inhibit DNA end resection in $G_1$ and $G_2$ phases of the cell cycle, and in S phase in mammalian cells (*Lee et al., 1998*; *Barlow et al., 2008*; *Clerici et al., 2008*; *Shao et al., 2012*).

While nuclease activity is largely limited in $G_0$/$G_1$ phase cells to prevent aberrant DNA end resection, evidence exists suggesting that nuclease-mediated DNA end processing occurs at some DSBs in $G_0$/$G_1$. For example, Artemis is required to open hairpin-sealed DNA ends generated during V(D) J recombination in lymphocytes (*Menon and Povirk, 2016*). Additionally, DNA end resection has been observed in $G_1$ phase after DNA damage at complex DNA lesions (*Averbeck et al., 2014*; *Biehs et al., 2017*), suggesting that DNA end resection is not completely inhibited in the absence of sister chromatids. Moreover, though CtIP phosphorylation by CDKs in $G_2$ is required for its activity during HR, CtIP also functions in $G_1$ at DSBs after phosphorylation by PLK3 (*Barton et al., 2014*). To investigate what additional factors may regulate DNA end resection in cells lacking sister chromatids, we performed a genome-wide CRISPR/Cas9 screen for genes whose inactivation either increases or decreases RPA bound to chromatin after irradiation (IR) in $G_0$-arrested murine cells. We discovered, unexpectedly, that KU70, KU80, and DNA-PKcs promote extensive DNA end resection in $G_0$ cells, but not in $G_1$ or $G_2$ phases of the cell cycle.

## Results

### RPA associates with IR-induced DNA DSBs in $G_0$ cells

Murine pre-B cells transformed with Abelson murine leukemia virus (termed abl pre-B cells hereafter) continuously proliferate in vitro and can be efficiently arrested in $G_0$, also referred to as the quiescent state, upon treatment with the abl kinase inhibitor imatinib (*Figure 1—figure supplement 1A*).

(*Bredemeyer et al., 2006*; *Chen et al., 2021a*). To investigate how DNA end resection is regulated in $G_0$ cells, we used a flow cytometric approach to assay RPA bound to chromatin after detergent extraction of soluble RPA, as a proxy for ssDNA generated at DSBs after exposing cells to irradiation (IR) (*Forment et al., 2012*; *Chen et al., 2021a*). This assay was performed in murine abl pre-B cell lines deficient in DNA Ligase IV (*Lig4*$^{-/-}$), to maximize our ability to detect chromatin-bound RPA at DSBs, given that completion of NHEJ is prevented in the absence of DNA Ligase IV. We also performed the analysis in *Lig4*$^{-/-}$:*Trp53bp1*$^{-/-}$ abl pre-B cells which lack the DNA end protection protein 53BP1 and accumulate high levels of RPA on chromatin after IR (*Chen et al., 2021a*). In agreement with our previous work, we detected a high level of chromatin-bound RPA in $G_0$-arrested *Lig4*$^{-/-}$:*Trp53bp1*$^{-/-}$ abl pre-B cells after IR, consistent with the role of 53BP1 in DNA end protection (*Figure 1A*). Surprisingly, we also observed RPA associated with chromatin after IR of $G_0$-arrested *Lig4*$^{-/-}$ abl pre-B cells, although at lower levels than in *Lig4*$^{-/-}$:*Trp53bp1*$^{-/-}$ abl pre-B cells (*Figure 1A*). Moreover, the increase in IR-induced chromatin-bound RPA does not require DNA Ligase IV deficiency as we were able to observe similar results using the RPA flow cytometric assay in wild-type (WT) murine abl pre-B cells arrested in $G_0$ (*Figure 1—figure supplement 1B*). These data indicate that extensive DNA end resection occurs at DSBs in $G_0$ cells, despite the presence of the DNA end protection proteins 53BP1 and KU70/KU80.

To determine whether higher levels of chromatin-bound RPA in irradiated $G_0$-arrested *Lig4*$^{-/-}$ abl pre-B cells is a result of DNA end resection, we depleted the nucleases that are required for the initiation of DNA end resection during HR in cycling cells. We found that the depletion of CtIP or MRE11 reduced the levels of RPA on chromatin in irradiated $G_0$-arrested *Lig4*$^{-/-}$ abl pre-B cells (*Figure 1B* and *Figure 1—figure supplement 1C*), indicating that the RPA we observe with our flow cytometric assay after IR is indeed a result of DNA end resection. Next, we investigated whether this observed chromatin-bound RPA depended on the nuclease Artemis, which has been shown to have endo and exonuclease activity and is essential in opening DNA hairpins during V(D)J recombination, which occurs in pre-B cells (*Ma et al., 2002*; *Ma et al., 2005*). We found that Artemis depletion had no effect on levels of RPA on chromatin in irradiated $G_0$-arrested wildtype abl pre-B cells, indicating that Artemis activity does not contribute to this process (*Figure 1—figure supplement 1D*).

To determine whether the DNA end resection that we observed was unique to murine abl pre-B cells or not, we performed the RPA flow cytometric chromatin association assay in the human breast epithelial cell line MCF10A. We arrested the MCF10A cells in $G_0$ by EGF deprivation (*Chen et al., 2021a*). Similarly, to *Lig4*$^{-/-}$ and WT murine abl pre-B cells in $G_0$, we observed IR-induced chromatin-bound RPA in $G_0$ human MCF10A cells (*Figure 1—figure supplement 1E*), consistent with DNA end resection occurring in these cells at DSBs. RPA binding to ssDNA surrounding DSBs often form distinct nuclear foci that can be easily detected by immunofluorescence staining and microscopy analysis (*Golub et al., 1998*). Therefore, we performed immunofluorescence staining for RPA in EGF-deprived MCF10A cells. We observed discrete IR-induced RPA foci, consistent with the RPA associated with ssDNA accumulating at DNA damage sites (*Figure 1C*). Together, these results suggest that broken DNA ends are resected in a CtIP and MRE11-dependent manner, leading to RPA accumulation on ssDNA in $G_0$ murine and human cells.

## DNA end resection and RPA loading occurs at site-specific DSBs in $G_0$ cells

As irradiation induces DNA base lesions and single-stranded DNA breaks in addition to DSBs, it could potentially complicate our analysis of DNA end processing at regions surrounding DSBs. Therefore, we investigated DSBs at specific locations in the mouse genome upon induction of the *AsiSI* endonuclease. We performed RPA chromatin immunoprecipitation sequencing (RPA ChIP-seq) after induction of *AsiSI* DSBs in $G_0$-arrested *Lig4*$^{-/-}$ murine abl pre-B cells. We detected RPA binding adjacent to *AsiSI* DSBs, consistent with ssDNA generated by resection around DNA DSBs (*Paiano et al., 2021*; *Figure 1D* and *Figure 1—figure supplement 1F*). Moreover, the association of RPA with chromatin was strand specific around the DSBs, consistent with the 5'–3' nature of DNA end resection which generates 3' ssDNA overhangs (*Paiano et al., 2021*; *Figure 1D*). To determine the extent of DNA end processing in $G_0$ cells, we performed END-seq (*Canela et al., 2016*; *Wong et al., 2021*) to directly measure DNA end resection at nucleotide resolution at *AsiSI*-induced DSBs, the majority of which are within 2 kb of the transcriptional start site of transcriptionally active genes (*Figure 1—figure supplement 2*). Using END-seq, we detected extensive DNA end resection in $G_0$-arrested *Lig4*$^{-/-}$ abl

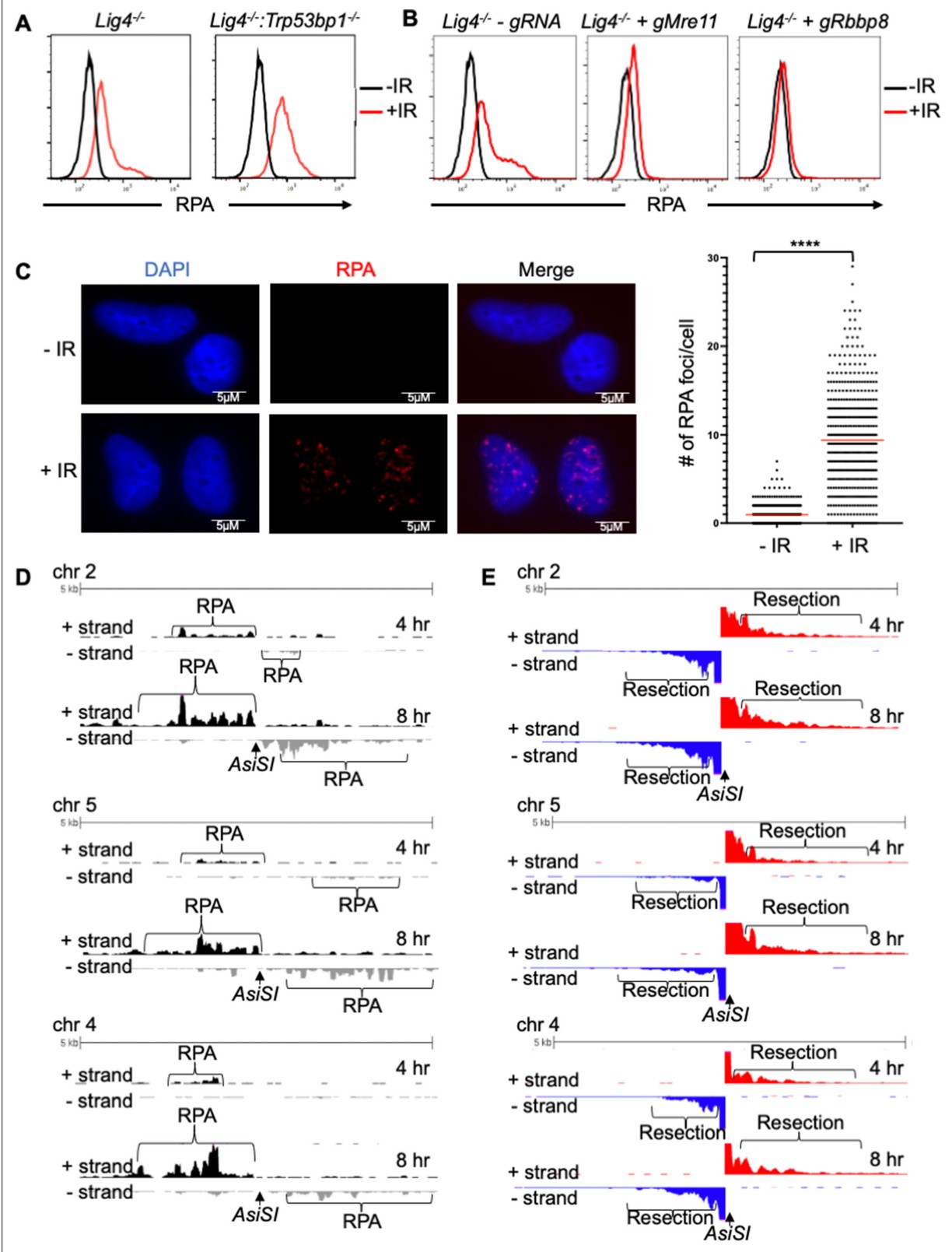

**Figure 1.** RPA is loaded onto ssDNA after DSBs in $G_0$ mammalian cells. (**A**) Flow cytometric analysis of chromatin-bound RPA in $G_0$-arrested *Lig4*[-/-] and *Lig4*[-/-]*:Trp53bp1*[-/-] abl pre-B cells before and 3 hr after 20 Gray IR. Representative of three independent experiments. (**B**) Flow cytometric analysis of chromatin-bound RPA before and 2 hr after 15 Gy IR in $G_0$-arrested *Lig4*[-/-] abl pre-B cells (left), *Lig4*[-/-] cells depleted of MRE11 (middle), and *Lig4*[-/-] cells depleted of CtIP (right). Representative of three independent experiments. (**C**) Representative images and quantification of IR-induced RPA

*Figure 1 continued on next page*

Figure 1 continued

foci from three independent experiments in $G_0$-arrested MCF10A cells before and 3 hr after 10 Gray IR. n=365 cells in -IR and n=433 cells in +IR. Red bars indicate average number of RPA foci in - IR = 0.96 and average number of RPA foci in +IR = 9.4 (****p<0.0001, unpaired t test). (**D**) RPA ChIP-seq tracks at *AsiSI* DSBs on chromosome 2, 5, and 4 at 4 hr (top) and 8 hr (bottom) after *AsiSI* endonuclease induction in $G_0$-arrested *Lig4$^{-/-}$* abl pre-B cells. (**E**) Representative END-Seq tracks showing resection at *AsiSI* DSBs at chromosome 2, 5, and 4 at 4 hr (top) and 8 hr (bottom) after *AsiSI* induction in $G_0$-arrested *Lig4$^{-/-}$* abl pre-B cells. END-seq data is representative from two independent experiments.

The online version of this article includes the following source data and figure supplement(s) for figure 1:

**Figure supplement 1.** RPA is loaded onto ssDNA after DSBs in $G_0$ mammalian cells.

**Figure supplement 1—source data 1.** Original western blots for **Figure 1**.

**Figure supplement 2.** The vast majority of *AsiSI* sites that are cleaved in $G_0$ cells are in close proximity to the transcription start site of actively transcribed genes.

pre-B cells at 4 and 8 hr after *AsiSI* DSB induction (**Figure 1E**). Together, these data indicate that in $G_0$-arrested cells, DNA ends are resected at DSBs induced by IR or site-specific endonucleases, generating ssDNA that is bound by RPA.

## A CRISPR/Cas9 screen identifies the DNA-PK complex as promoting DNA end resection in $G_0$ cells

To identify factors that influence DNA end resection in $G_0$ cells, we performed a genome-wide CRISPR/Cas9 screen in $G_0$-arrested *Lig4$^{-/-}$* murine abl pre-B cells 2 hr after irradiation to identify factors that either promote or impair DNA end resection (**Figure 2A**). We isolated the 10% of cells with the lowest RPA (low RPA) and the 10% cells with the highest RPA (high RPA) staining intensity using our RPA flow cytometric assay followed by flow assisted cell sorting. We then amplified the guide RNAs (gRNAs) in these populations of cells and determined their frequencies using high-throughput sequencing. gRNAs enriched in the low RPA staining population correspond to genes encoding proteins that normally promote DNA end resection, while gRNAs enriched in the high RPA population correspond to genes encoding proteins that normally impair resection. In this screen, we identified several gRNAs enriched in the low RPA staining population to *Rbbp8* which encodes the nuclease CtIP, and *Nbn*, which encodes the NBN subunit of the MRE11-RAD50-NBN (MRN) complex, consistent with their established roles in promoting DNA end resection (**Figure 2B**). Unexpectedly, we also found gRNAs targeting Xrcc6 (the gene encoding KU70), *Xrcc5* (the gene encoding KU80), and *Prkdc* (the gene encoding DNA-PKcs) highly enriched in our low RPA population (**Figure 2B**). This suggested that DNA-PK may promote DNA end resection in $G_0$ cells, contrary to the established role of these factors in preventing DNA end resection in other phases of the cell cycle.

To validate the screen and determine if DNA-PK is required for DNA end resection, we generated *Lig4$^{-/-}$:Prkdc$^{-/-}$* abl pre-B cells that do not express DNA-PKcs by CRISPR/Cas9-mediated gene inactivation (**Figure 2—figure supplement 1A**). $G_0$-arrested *Lig4$^{-/-}$:Prkdc$^{-/-}$* abl pre-B cells had lower levels of chromatin-bound RPA after IR compared to *Lig4$^{-/-}$* abl pre-B cells (**Figure 2C**). DNA-PKcs and Ataxia-telangiectasia mutated (ATM) are two major serine/threonine kinases that are activated in response to DNA DSBs and share some overlapping functions due to similar substrate specificity (**Blackford and Jackson, 2017**). Because DNA-PKcs but not ATM was identified in our screen, we wanted to determine if the pro-resection activity in $G_0$-arrested cells is unique to DNA-PKcs or also shared by ATM. We treated $G_0$-arrested *Lig4$^{-/-}$* abl pre-B cells with the ATM inhibitor KU55933 or the DNA-PK inhibitor NU7441 before IR and performed flow cytometric analysis of IR-induced chromatin-bound RPA. In contrast to the consistent reduction in the levels of chromatin-bound RPA observed in $G_0$-arrested *Lig4$^{-/-}$* abl pre-B cells treated with DNA-PK inhibitor, ATM inhibition did not have a detectable effect on the levels of IR-induced binding of RPA in $G_0$-arrested *Lig4$^{-/-}$* abl pre-B cells (**Figure 2D** and **Figure 2—figure supplement 1B**). Additionally, DNA-PK inhibition in wild type abl pre-B cells arrested in $G_0$ showed a modest effect in reducing levels of chromatin-bound RPA (**Figure 2—figure supplement 1C**). The role of DNA-PK in promoting DNA end resection in $G_0$ is not limited to murine abl pre-B cells as we also observed a reduced number of IR-induced RPA foci in $G_0$-arrested human MCF10A cells upon inhibition of DNA-PK (**Figure 2—figure supplement 1D**). These results indicate that DNA-PKcs activity, but not ATM, uniquely promotes resection and RPA binding to damaged chromatin after IR in $G_0$ cells.

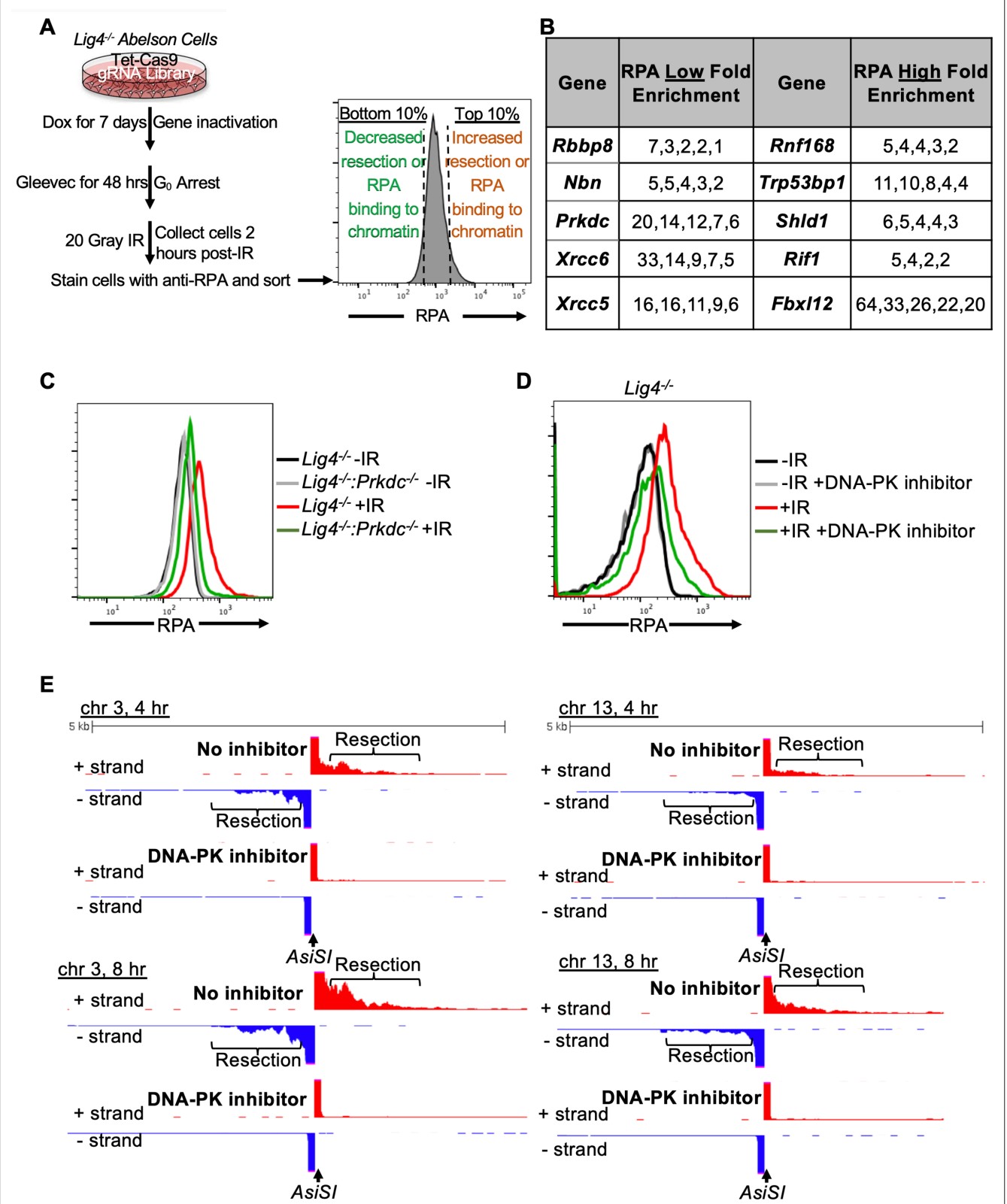

**Figure 2.** A genome-wide gRNA screen identifies DNA-PK as a factor that promotes DNA end resection in $G_0$. (**A**) Schematic of a genome-wide gRNA screen for factors promoting (bottom 10%/RPA low) or inhibiting (top 10%/RPA high) chromatin-bound RPA loading 2 hr after 20 Gray IR in $G_0$-arrested $Lig4^{-/-}$ abl pre-B cells. (**B**) Fold enrichment of selected gRNAs in low RPA and high RPA populations. Fold enrichment was calculated as the ratio of normalized read number of gRNAs in the low RPA population and that in the high RPA population and vice versa (n=1). (**C**) Flow cytometric

*Figure 2 continued on next page*

*Figure 2 continued*

analysis of chromatin-bound RPA in $G_0$-arrested *Lig4$^{-/-}$* and *Lig4$^{-/-}$:Prkdc$^{-/-}$* abl pre-B cells before and 3 hr after 15 Gray IR. Data is representative of three independent experiments in two different cell lines. (**D**) Flow cytometric analysis of chromatin-bound RPA in $G_0$-arrested *Lig4$^{-/-}$* abl pre-B cells with and without 10 μM NU7441 (DNA-PK inhibitor) pre-treatment 1 hr before 20 Gray IR. Data is representative of three independent experiments in two different cell lines. (**E**) Representative END-seq tracks at chromosome 3 (left) and chromosome 13 (right) in $G_0$-arrested *Lig4$^{-/-}$* abl pre-B cells 4 hr (top) and 8 hr (bottom) after *AsiSI* DSB induction, with and without 10 μM NU7441 treatment.

The online version of this article includes the following source data and figure supplement(s) for figure 2:

**Figure supplement 1.** A genome-wide gRNA screen identifies DNA-PK as a factor that promotes DNA end resection in $G_0$.

**Figure supplement 1—source data 1.** Original western blots for *Figure 2*.

To directly observe if DNA-PKcs influenced DNA end resection at DSBs, we performed nucleotide resolution END-seq on $G_0$-arrested *Lig4$^{-/-}$* murine abl pre-B cells with and without DNA-PK inhibitor treatment before the induction of *AsiSI* DSBs. Consistent with our RPA flow cytometric assay results, DNA-PK inhibitor-treated $G_0$-arrested *Lig4-/-* abl pre-B cells showed greatly reduced END-Seq signals distal to DSBs, consistent with limited DNA end processing when DNA-PK is inactivated (*Figure 2E* and *Figure 2—figure supplement 1E*). These results demonstrate that DNA-PK activity promotes DNA end resection of DSBs in $G_0$ mammalian cells.

## FBXL12 inhibits KU70/KU80-dependent DNA end resection in $G_0$ cells

Given that DNA-PKcs promotes DNA end resection in $G_0$ cells (*Figure 2C, D and E*, *Figure 2—figure supplement 1C and D*, 1E), and that *Xrcc6* and *Xrcc5* (genes encoding KU70 and KU80) were enriched in the RPA low population of cells in the CRISPR/Cas9 screen (*Figure 2B*), we determined whether KU70/KU80 may also promote resection in $G_0$ cells. We generated *Lig4$^{-/-}$:Xrcc6$^{-/-}$* murine abl pre-B cells and measured DNA end resection using our RPA flow cytometric approach. Consistent with our observations in DNA-PK inhibitor-treated $G_0$-arrested *Lig4$^{-/-}$* abl pre-B cells and *Lig4$^{-/-}$:Prkdc$^{-/-}$* abl pre-B cells, the level of chromatin-bound RPA after IR was greatly reduced in $G_0$-arrested *Lig4$^{-/-}$:Xrcc6$^{-/-}$* abl pre-B cells compared to *Lig4$^{-/-}$* abl pre-B cells (*Figure 3A* and *Figure 3—figure supplement 1A*). As such, the entire DNA-PK complex is required for DNA end resection in $G_0$ cells.

KU70/KU80 is removed from DSBs via ubiquitylation, which has been shown to be mediated by E3 ligases including RNF138, RNF8, RNF126, and the SCF$^{Fbxl12}$ complex (*Postow et al., 2008*; *Feng and Chen, 2012*; *Postow and Funabiki, 2013*; *Ismail et al., 2015*; *Ishida et al., 2017*). In agreement, gRNAs targeting *Fbxl12*, which encodes the substrate recognition subunit FBXL12 of the SCF$^{Fbxl12}$ E3 ubiquitin ligase complex, were highly enriched in our screen in the high RPA staining cell population (*Figure 2B*), consistent with the idea that the persistent presence of KU70/KU80 at DSBs in cells lacking FBXL12 would lead to extensive DNA end resection. Indeed, we observed that in $G_0$-arrested *Lig4$^{-/-}$:Fbxl12$^{-/-}$* murine abl pre-B cells, the level of IR-induced chromatin-bound RPA increased compared to *Lig4$^{-/-}$* abl pre-B cells (*Figure 3B*). Given the role of FBXL12 on limiting the levels of KU70/KU80 at broken DNA ends, we tested whether the increased DNA end resection phenotype in *Lig4$^{-/-}$:Fbxl12$^{-/-}$* abl pre-B cells depended on DNA-PK activity or the presence of the KU70/KU80 complex. Indeed, inhibition of DNA-PK with NU7441 (*Figure 3C*) and depletion of KU70 (*Figure 3D* and *Figure 3—figure supplement 1B*) in $G_0$-arrested *Lig4$^{-/-}$:Fbxl12$^{-/-}$* abl pre-B cells prevented excessive accumulation of RPA on chromatin after IR. Our results suggest that the ability of DNA-PK to promote DNA end resection in $G_0$ cells is regulated through maintaining proper levels of KU70/KU80 at DNA DSBs by the SCF$^{Fbxl12}$ E3 ubiquitylation complex.

## DNA-PK uniquely promotes DNA end resection exclusively in $G_0$ cells

KU70/KU80 have been shown to prevent DNA end resection in $G_1$ and $G_2$ phases in budding yeast and in S phase in mammalian cells but has not been examined in $G_0$ cells (*Lee et al., 1998*; *Clerici et al., 2008*; *Shao et al., 2012*). Thus, we set out to determine whether DNA-PK-dependent DNA end resection is limited to $G_0$ or can occur in other phases of the cell cycle. To this end, we compared the levels of IR-induced chromatin bound RPA in *Lig4$^{-/-}$*, *Lig4$^{-/-}$:Prkdc$^{-/-}$* and *Lig4$^{-/-}$:Xrcc6$^{-/-}$* murine abl pre-B cells arrested in $G_0$ by imatinib, arrested in $G_2$ by the CDK1 inhibitor RO3306, and in $G_1$ phase (cells with 2 N DNA) in a proliferating population. In contrast to $G_0$ cells, loss of DNA-PKcs (*Lig4$^{-/-}$:Prkdc$^{-/-}$*) did not reduce the levels of IR-induced chromatin-bound RPA in $G_2$-arrested or cycling $G_1$ phase cells

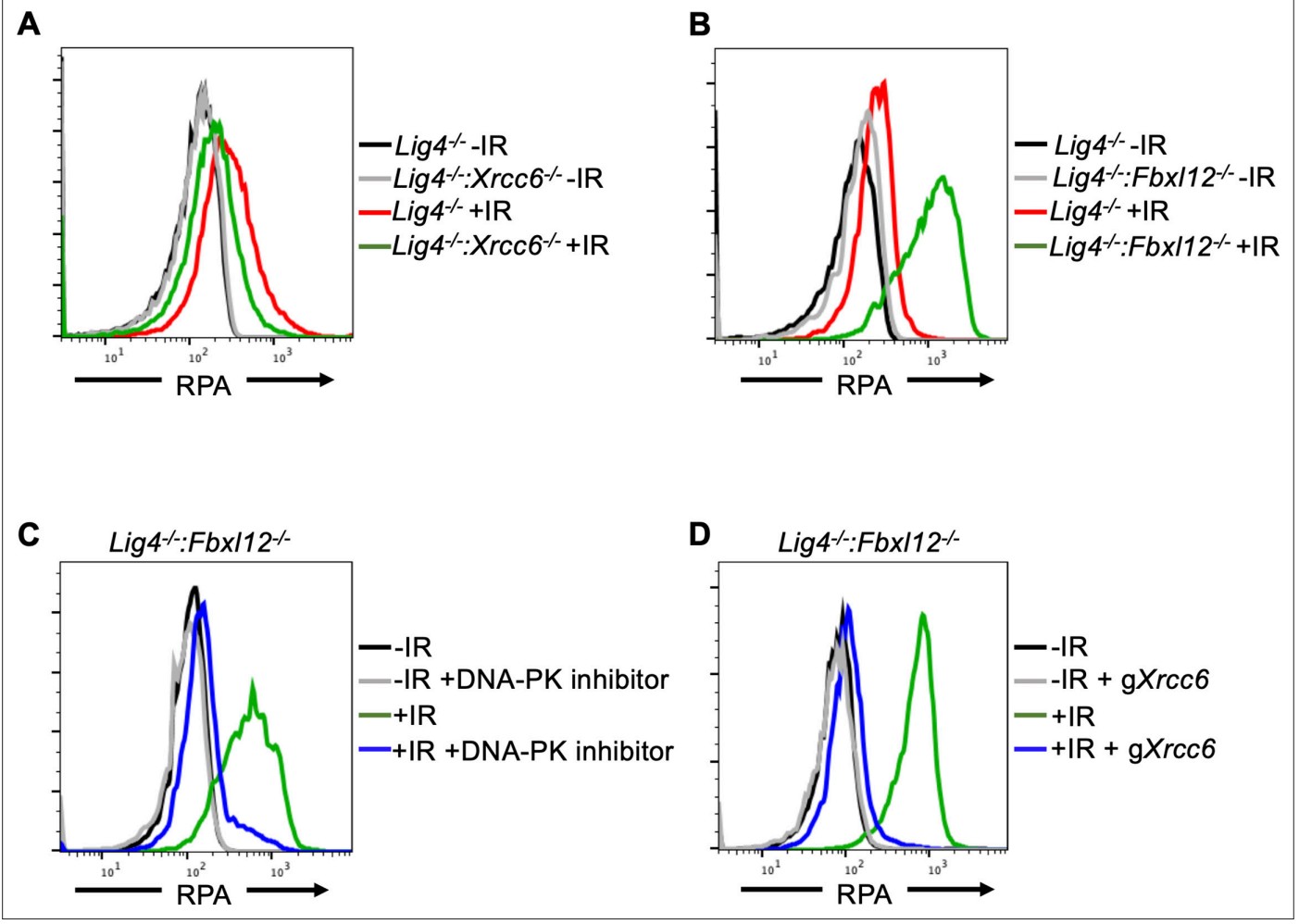

**Figure 3.** FBXL12 inhibits KU70/KU80-promoted DNA end resection. (**A**) Flow cytometric analysis of chromatin-bound RPA in $G_0$-arrested $Lig4^{-/-}$ abl pre-B cells and $Lig4^{-/-}:Xrcc6^{-/-}$ abl pre-B cells before and 3 hr after 20 Gray IR. Data is representative of three independent experiments in two different cell lines. (**B**) As in A, in $G_0$-arrested $Lig4^{-/-}$ and $Lig4^{-/-}:Fbxl12^{-/-}$ abl pre-B cells. Data is representative of three independent experiments in at least two different cell lines. (**C**) Flow cytometric analysis of chromatin-bound RPA in $G_0$-arrested $Lig4^{-/-}:Fbxl12^{-/-}$ abl pre-B cells with and without 10 µM NU7441 treatment, before and 3 hr after 20 Gray IR. Data is representative of three independent experiments in at least two different cell lines (**D**) Flow cytometric analysis of chromatin-bound RPA in $G_0$-arrested $Lig4^{-/-}:Fbxl12^{-/-}$ abl pre-B cells before and after $Xrcc6$ knockout, before and 3 hr after 15 Gray IR. Data is representative of three independent experiments.

The online version of this article includes the following source data and figure supplement(s) for figure 3:

**Figure supplement 1.** FBXL12 inhibits KU70/KU80-promoted DNA end resection.

**Figure supplement 1—source data 1.** Original western blots showing Ku70 depletion in **Figure 3—figure supplement 1**.

**Figure supplement 1—source data 2.** Original western blots for Ku70 depletion in Fbxl12- cells.

(**Figure 4A** and **Figure 4—figure supplement 1A**). Similar results were obtained when analyzing $Lig4^{-/-}:Xrcc6^{-/-}$ abl pre-B cells (**Figure 4B**). The unique function of DNA-PK activity in promoting DNA end resection in $G_0$-arrested cells was confirmed with END-seq analysis of *AsiSI*-induced DSBs in $Lig4^{-/-}$ abl pre-B cells arrested in $G_0$ or $G_2$ and treated with or without DNA-PK inhibitor. Whereas $G_0$-arrested $Lig4^{-/-}$ abl pre-B cells treated with DNA-PK inhibitor exhibited significantly reduced END-seq signals in regions distal to the DSBs, the same treatment had little effect in cells arrested in $G_2$ phase of the cell cycle (**Figure 4C** and **Figure 4—figure supplement 1B**). Quantitation of the resection tract lengths from the End-seq analysis showed that they were on average 3–4 kb in the $G_0$-arrested cells and were greatly reduced upon treatment with DNA-PK inhibitor (**Figure 4D**). In comparison, the resection tract lengths in $G_2$ arrested cells were minimally affected by treatment with DNA-PK inhibitor (**Figure 4D**). Additionally, NHEJ-proficient wild-type MCF10A cells arrested in $G_0$, but not cells in

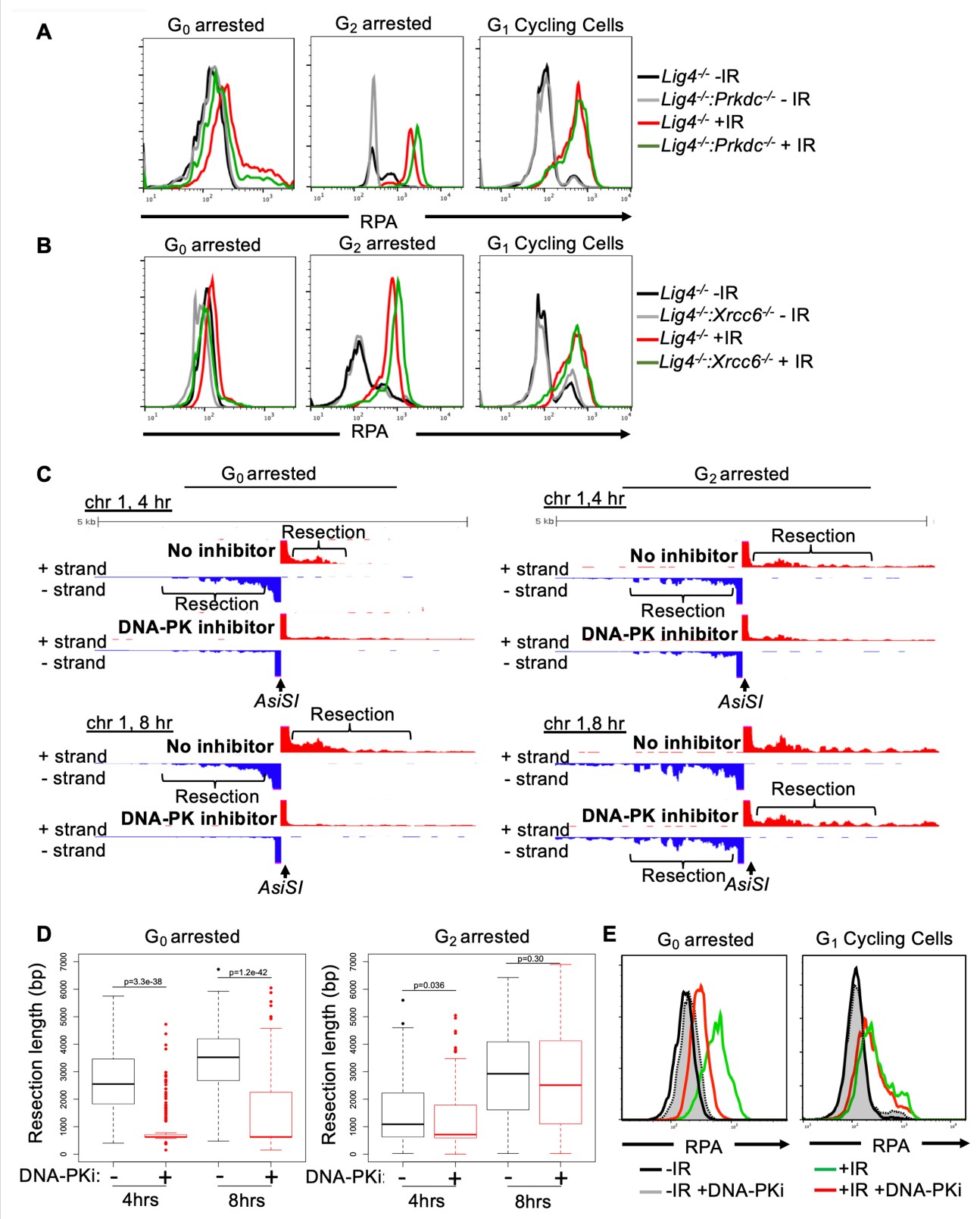

**Figure 4.** DNA-PK mediates DNA end resection in $G_0$ but not in $G_1$ or $G_2$ phases of the cell cycle. (**A**) Flow cytometric analysis of chromatin-bound RPA in *Lig4*[-/-] and *Lig4*[-/-]:*Prkdc*[-/-] abl pre-B cells arrested in $G_0$ (left), arrested in $G_2$ by 10 μM RO-3306 treatment for 16 hr and gated on 4 N (middle), and $G_1$ cells gated on 2 N DNA content in cycling cells (right), before and 3 hr after 20 Gray IR. Data is representative of three independent experiments in at least two different cell lines. (**B**) As in A in *Lig4*[-/-] and *Lig4*[-/-]:*Xrcc6*[-/-] abl pre-B cells. (**C**) Representative END-seq tracks in $G_0$ (left) and $G_2$-arrested (right,

*Figure 4 continued on next page*

*Figure 4 continued*

by 10 μM RO-3306 treatment for 16 hr) *Lig4*[-/-] abl pre-B cells, with and without 10 μM NU7441 treatment on chromosome 1, 4 hr (top) and 8 hr (bottom) after *AsiSI* endonuclease induction. (**D**) Average resection length in $G_0$-arrested *Lig4*[-/-] abl pre-B (left) and $G_2$-arrested *Lig4*[-/-] abl pre-B (right) 4 and 8 hr after *AsiSI* DSB induction, with and without 10 μM NU7441 treatment (DNA-PKi). (**E**) Flow cytometric analysis of chromatin-bound RPA 4 hr after 20 Gray IR in MCF10A cells arrested in $G_0$ after EGF withdrawal for 48 hr or cycling cells gated on 2 N DNA content, with and without 10 μM NU7441 treatment (DNA-PKi).

The online version of this article includes the following figure supplement(s) for figure 4:

**Figure supplement 1.** DNA-PK mediates DNA end resection in $G_0$ but not $G_2$.

$G_1$ phase, exhibited reduced RPA on chromatin after IR upon DNA-PK inhibition (*Figure 4E*). These results suggest that DNA-PK distinctly promotes DNA end resection at DSBs in mammalian cells in $G_0$ but not in other cell cycle phases.

## Discussion

DNA end resection is one of the key events that determines whether cells utilize NHEJ, HR, or other repair pathways utilizing homologous sequences. During $G_0$ and $G_1$ phase of the cell cycle, NHEJ is the predominant DSB repair pathway and DNA end resection is largely limited compared to other phases of the cell cycle. However, in this study we revealed that DNA end resection dependent on CtIP and MRE11, which are required for resection in S and $G_2$ phases of the cell cycle, occurs at DSBs in $G_0$ mammalian cells (*Figure 1B*). Because CtIP activity in $G_1$, $G_2$ and S phases requires its phosphorylation, this is likely to be the case in $G_0$ cells and future studies will identify the kinase responsible for any CtIP phosphorylation in $G_0$ cells. In addition to CtIP and MRE11, we identified additional factors that promote resection in $G_0$ cells as components of the DNA-PK complex, including KU70, KU80, and DNA-PKcs, in a genome-wide CRISPR/Cas9 screen and showed that the kinase activity of DNA-PK is

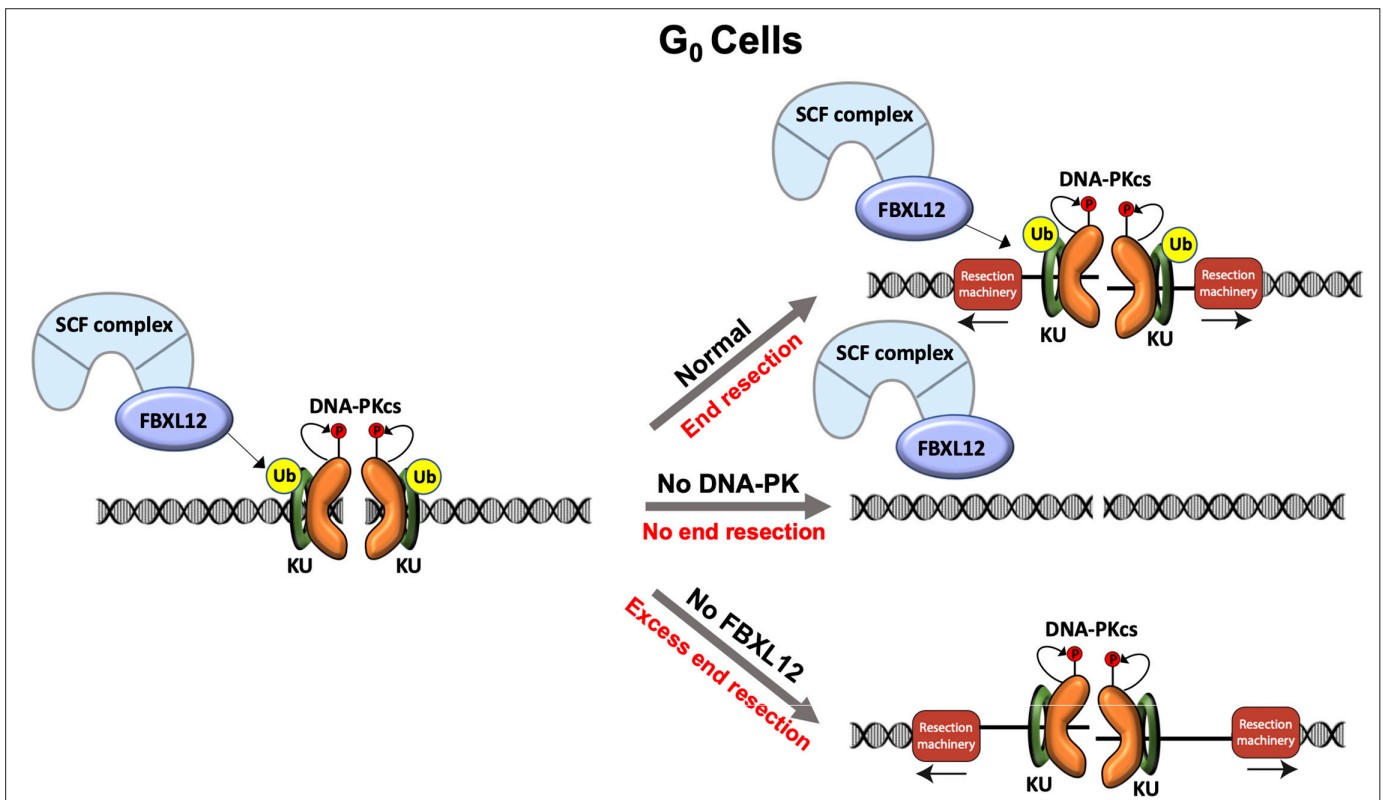

**Figure 5.** Model of DNA-PK-mediated DNA end resection in $G_0$ cells. Normally in $G_0$ phase at DSBs, the DNA-PK complex promotes DNA end resection. This resection is counteracted by FBXL12. Without DNA-PK, there is no DNA end resection in $G_0$. Without FBXL12, DNA-PK persists at DSBs and leads to more extensive DNA end resection.

critical as resection of DSBs diminishes upon DNA-PK inhibitor treatment (*Figures 2 and 3*). Interestingly, we also found in our genome wide CRISPR/Cas9 screen that inactivating FBXL12, the substrate recognition subunit of the SCF$^{FBXL12}$ E3 ubiquitin ligase complex, promotes extensive resection of DNA ends in $G_0$ cells (*Figure 3B*). As the SCF$^{FBXL12}$ E3 ubiquitin is thought to limit the abundance of the KU70/KU80 heterodimer (*Postow and Funabiki, 2013*), our data are in line with the notion that loss of FBXL12 results in aberrant accumulation of KU70/KU80 at DSBs, and consequently elevated or prolonged activation of DNA-PK at DSBs which promotes resection in $G_0$ cells (*Figure 5*).

Why would resection occur in $G_0$ cells? Chemical modifications or secondary structures at DSBs have been identified as requiring DNA end processing to create a more accessible repair environment, which could presumably be the case at DSBs in $G_0$ cells (*Weinfeld and Soderlind, 1991*). For example, Artemis is an endo and exonuclease which is activated by DNA-PKcs and uses its nuclease activity to open DNA hairpins at coding ends, which is required for V(D)J recombination, and cleaves 3' ssDNA overhangs during NHEJ (*Ma et al., 2002*; *Ma et al., 2005*). Artemis was also recently shown to contribute to slow, resection dependent NHEJ repair in $G_1$ phase cells (*Biehs et al., 2017*). Though Artemis does not have a role in DNA end resection in $G_0$ (*Figure 1—figure supplement 1D*). it serves as an example of nuclease activity being critical for DSB repair outside of HR. It is additionally possible that DNA end resection in $G_0$ results in substrates that are ideal for Pol Theta-mediated end joining (TMEJ). TMEJ occurs after extensive DNA end resection when HR is not possible or when substrates are not suitable for NHEJ, which could be the case at a subset of breaks in $G_0$ (*Yousefzadeh et al., 2014*; *Wyatt et al., 2016*). However, it is notably that TMEJ is KU70/KU80 independent, while the resection that we see in $G_0$ is KU70/KU80 dependent. Interestingly, DNA end resection has a role in recruiting anti-resection factors to limit extensive DNA end resection. The SHLD2 component of Shieldin binds ssDNA, suppresses RAD51 loading, and ultimately limits DNA end resection by preventing access to resection nucleases (*Noordermeer et al., 2018*). HELB, a 5'–3' DNA helicase, binds to RPA and limits EXO1 and BLM-DNA2-mediated DNA end resection (*Tkáč et al., 2016*). In this way, limited DNA end resection in $G_0$ cells could be important in preventing more extensive DNA end resection. Altogether, we propose that DNA end resection in $G_0$ cells is likely not resulting in aberrant HR but may be required to create more accessible DNA ends and/or to recruit anti-resection factors.

Studies investigating the role of KU70/KU80 during DSB repair have found that KU70/KU80 protects DSBs from nuclease activity. For example, at HO endonuclease breaks in budding yeast, deletion of KU70/KU80 leads to ssDNA accumulation in $G_1$ cells and increased MRE11 recruitment to DSBs compared to wild-type cells (*Lee et al., 1998*; *Clerici et al., 2008*). Also in budding yeast, at inducible I-SceI DSBs, deletion of KU70 results in increased RFA1 foci formation in $G_1$, but deletion of NHEJ factor DNA Ligase IV leads to no defect in RFA1 foci formation compared to wild-type cells, indicating that KU70 itself, not NHEJ, is a barrier to DNA end resection (*Barlow et al., 2008*). In mammalian cells, complementation of KU70/KU80 knockout cells with a *M. tuberculosis* KU homolog persistently bound to DSBs in S phase results in reduced RPA and RAD51 foci formation after IR (*Shao et al., 2012*). Contrary to these roles for KU70/KU80 in protecting DNA ends from nucleolytic attack, we found that in $G_0$ cells, KU70/KU80 promotes DNA end resection (*Figures 3A and 4B*). We hypothesize that KU70/KU80 promotes resection through recruitment and activation of DNA-PKcs at DSBs (*Gottlieb and Jackson, 1993*), as we also found that DNA-PKcs inhibition and genetic deletion of *Prkdc* leads to less RPA on chromatin after IR and shorter tracts of DNA end resection in $G_0$ cells (*Figure 2C–E*, *Figure 2—figure supplement 1C and D*, 4A, 4C, 4D). It is important to note that most studies establishing the role of KU70/KU80 in protecting DNA ends were performed in *S. cerevisiae* which do not have a homolog to DNA-PKcs. Therefore, we hypothesize that the function of DNA-PK promoting DNA end resection in $G_0$ cells may not be evolutionarily conserved. Moreover, previous studies in *S. cerevisiae* and mammalian cells establishing DNA-PK as a pro-NHEJ complex did not analyze $G_0$ cells. We found that DNA-PK does not promote DNA end resection in $G_1$ or $G_2$ phase cells, only in $G_0$-arrested cells, indicating that DNA-PK-dependent DNA end resection is unique to $G_0$, but is not contradictory to its anti-resection function in $G_1$ or $G_2$ phase cells (*Figure 4*). In $G_0$ cells, KU70/KU80 could protect some DNA ends, but after recruitment and activation of DNA-PKcs, the net effect is DNA end resection. Additional studies may elucidate how the balance between DNA end protection and DNA end resection is regulated in $G_0$.

ATM and DNA-PK have been shown to have some overlapping functions in DNA damage response and repair, including phosphorylation of H2A.X in response to IR and signal join formation during V(D)

J recombination (*Stiff et al., 2004*; *Gapud et al., 2011*; *Zha et al., 2011*). However, we find that this is not the case during DNA end resection in $G_0$ cells as DNA-PK promotes resection in $G_0$ cells, but ATM does not have a detectable impact (*Figure 2—figure supplement 1B*). ATM has been implicated in promoting HR repair by phosphorylating CtIP and promoting KU70/KU80 removal from DSBs, as well as phosphorylating DNA-PKcs at single-ended DSBs to remove it from these breaks that require DNA end resection (*Wang et al., 2013*; *Britton et al., 2020*). DNA-PKcs autophosphorylation promotes HR by removing it from DSBs to allow nuclease access but is typically associated with promoting NHEJ by phosphorylating Artemis, XRCC4, and XLF (*Zhou and Paull, 2013*; Bartlett and Lees-Miller 2018*Bartlett and Lees-Miller, 2018*). So while ATM often promotes DNA end resection and HR, it appears that DNA-PKcs could be acting in place of ATM to promote DNA end resection in $G_0$ cells. It is additionally possible that DNA-PKcs phosphorylates a unique substrate(s) in $G_0$ cells that promotes DNA end resection.

In summary, we provide here evidence that DNA-PK promotes DNA end resection uniquely in $G_0$ cells, and that this DNA end resection is counteracted by FBXL12. We speculate that some aspects of DSB repair in $G_0$ function differently than DSB repair in cycling cells, and future studies may reveal the mechanism and utility of these key differences.

# Materials and methods

## Key resources table

| Reagent type (species) or resource | Designation | Source or reference | Identifiers | Additional information |
|---|---|---|---|---|
| Antibody | Anti-CtIP (Rabbit polyclonal) | N/A | custom made (Richard Baer, Columbia University) | WB (1:1000) |
| Antibody | Anti-MRE11 (Rabbit polyclonal) | Novus Biologicals | NB100-142 RRID:AB_1109376 | WB (1:2000) |
| Antibody | Anti-GAPDH (GAPDH-71.1) (Mouse monoclonal) | Millipore Sigma | G8795 RRID:AB_1078991 | WB (1:10000) |
| Antibody | Anti-KAP1 (N3C2) (Rabbit polyclonal) | Genetex | GTX102226 RRID:AB_2037324 | WB (1:2000) |
| Antibody | Anti-RPA32 (4E4) (Rat monoclonal) | Cell Signaling Technology | 2,208 S RRID:AB_2238543 | WB (1:1000) FC (1:200) IF (1:500) |
| Antibody | Anti-KU70 (D10A7) (Rabbit monoclonal) | Cell Signaling Technology | 4,588 S RRID:AB_11179211 | WB (1:1000) |
| Antibody | Anti-DNA-PK (SC57-08) (Rabbit monoclonal) | Invitrogen | MA5-32192 RRID:AB_2809479 | WB (1:1000) |
| Antibody | Anti-RPA32 (rabbit polyclonal) | Abcam | ab10359 RRID:AB_297095 | ChIP (10 ug) |
| Antibody | HRP, goat anti-mouse (goat polyclonal) | Promega | W4021 RRID:AB_430834 | WB (1:5000) |
| Antibody | HRP, goat anti-rabbit IgG (goat polyclonal) | Promega | W4011 RRID:AB_430833 | WB (1:5000) |
| Antibody | Alexa Fluor 488, goat anti-rat IgG (goat polyclonal) | BioLegend | 405,418 RRID:AB_2563120 | FC (1:500) |
| Antibody | Alexa Fluor 647, goat anti-rat IgG (goat polyclonal) | BioLegend | 405,416 RRID:AB_2562967 | FC (1:500) |
| Antibody | Alexa Fluor 594, goat anti-rat IgG (goat polyclonal) | BioLegend | 405,422 RRID:AB_2563301 | IF (1:500) |
| Recombinant DNA reagent | pCW-Cas9 (plasmid) | Addgene | 50,661 RRID:Addgene_50661 | |
| Recombinant DNA reagent | pKLV-U6 gRNA(BbsI)-PGKpuro-2ABFP (plasmid) | Addgene | 50,946 RRID:Addgene_50946 | |
| Recombinant DNA reagent | Genome-wide CRISPR guide RNA library V2 (plasmid) | Addgene | 67,988 RRID:Addgene_67988 | |

*Continued on next page*

*Continued*

| Reagent type (species) or resource | Designation | Source or reference | Identifiers | Additional information |
|---|---|---|---|---|
| Cell line (*H. sapiens*) | *MCF10A* | ATCC | CRL-10317 RRID:CVCL_0598 | |
| Cell line (*H. sapiens*) | *MCF10A: iCas9* | This study | Clone 25 | Available upon request |
| Cell line (*M. musculus*) | WT:*iCas9* abl pre-B cells | This study | M63.1.MG36.iCas9.302 | Available upon request |
| Cell line (*M. musculus*) | *Lig4$^{-/-}$:iCas9* abl pre-B cells | This study | A5.83.MG9.iCas9.16 | Available upon request |
| Cell line (*M. musculus*) | *Lig4$^{-/-}$:iCas9* abl pre-B cells | This study | A5.115.iCas9.72 | Available upon request |
| Cell line (*M musculus*) | *Lig4$^{-/-}$:Trp53bp1:iCas9* abl pre-B cells | This study | Clone 82 | Available upon request |
| Cell line (*M musculus*) | *Lig4$^{-/-}$:Xrcc6$^{-/-}$:iCas9* abl pre-B cells | This study | Clones 134 and 140 | Available upon request |
| Cell line (*M. musculus*) | *Lig4$^{-/-}$:Prkdc$^{-/-}$:iCas9* abl pre-B cells | This study | Clone 6 | Available upon request |
| Cell line (*M. musculus*) | *Lig4$^{-/-}$:Fbxl12$^{-/-}$:iCas9* abl pre-B cells | This study | Clone 6 | Available upon request |
| Cell line (*M. musculus*) | *Lig4$^{-/-}$:iAsiSI* abl pre-B cells | This study | Clone 20 | Available upon request |
| Chemical compound, drug | Imatinib | Selleckchem | S2475 | |
| Chemical compound, drug | Doxycycline | Sigma-Aldrich | D9891 | |
| Chemical compound, drug | Polybrene | Sigma Aldrich | S2667 | |
| Chemical compound, drug | Lipofectamine 2000 | Thermo Fisher Scientific | 11668019 | |
| Chemical compound, drug | NU7441 | Selleck Chemicals | S2638 | |
| Chemical compound, drug | KU-55933 | Selleck Chemicals | S1092 | |
| Chemical compound, drug | EGF | PeproTech | AF-100–15 | |
| Chemical compound, drug | Hydrocortisone | Sigma-Aldrich | H-0888 | |
| Chemical compound, drug | Cholera Toxin | Sigma-Aldrich | C-8052 | |
| Chemical compound, drug | Insulin | Sigma-Aldrich | I-1882 | |
| Commercial assay, kit | 7-AAD (DNA stain) | BD Biosciences | 559,925 RRID:AB_2869266 | |
| Commercial assay, kit | Cytofix/Cytoperm solution | BD Biosciences | 554,722 RRID:AB_2869010 | |
| Commercial assay, kit | Perm/Wash Buffer | BD Biosciences | 554,723 RRID:AB_2869011 | |
| Commercial assay, kit | FITC BrdU Flow Kit | BD Biosciences | 559,619 RRID:AB_2617060 | |

*Continued on next page*

*Continued*

| Reagent type (species) or resource | Designation | Source or reference | Identifiers | Additional information |
|---|---|---|---|---|
| Sequence-based reagent | pKLV lib330F | This study designed based on [*Tzelepis et al., 2016*] | PCR primers | AATGGACTATCATATGCTTACCGT |
| Sequence-based reagent | pKLV lib490R | This study designed based on *Tzelepis et al., 2016* | PCR primers | CCTACCGGTGGATGTGGAATG |
| Sequence-based reagent | PE.P5_pKLV lib195 Fwd | This study designed based on *Tzelepis et al., 2016* and standard Illumina adaptor sequences | PCR primers | AATGATACGGCGACCACCGAGATCTGG CTTTATATATCTTGTGGAAAGGAC |
| Sequence-based reagent | P7 index180 Rev | This study designed based on *Tzelepis et al., 2016* and standard Illumina adaptor sequences | PCR primers | CAAGCAGAAGACGGCATACGAGAT *INDEX*GTGACTGGAGTTCAGACGTG TGCTCTTCCGATCCAGACTGCCTTGGGAAAAGC |
| Sequence-based reagent | BU1 | *Canela et al., 2016* | PCR primers | 5'-Phos-GATCGGAAGAGCGTCGT GTAGGGAAAGAGTGUU[Biotin-dT]U [Biotin-dT]UUACACTCTTTC CCTACA CGACGCTCTTCCGATC* T-3' [*phosphorothioate bond] |
| Sequence-based reagent | BU2 | *Canela et al., 2016* | PCR primers | 5'-Phos-GATCGGAAGAGCACACG TCUUUUUUUUUAGACGTGTGCTC TTCCGATC*T-3' [*phosphorothioate bond] |
| Sequence-based reagent | *Trp53bp1* gRNA sequence | Sequence is from *Tzelepis et al., 2016* | N/A | GAACCTGTCAGACCCGATC |
| Sequence-based reagent | *Rbbp8* gRNA sequence | Sequence is from *Tzelepis et al., 2016* | N/A | ATTAACCGGCTACGAAAGA |
| Sequence-based reagent | *Mre11 gRNA sequence* | Sequence is from *Tzelepis et al., 2016* | N/A | TGCCGTGGATACTAAATAC |
| Sequence-based reagent | *Prkdc gRNA sequence* | Sequence is from *Tzelepis et al., 2016* | N/A | ATGCGTCTTAGGTGATCGA |
| Sequence-based reagent | *Xrcc6 gRNA sequence* | Sequence is from *Tzelepis et al., 2016* | N/A | CCGAGACACGGTTGGCCAT |
| Sequence-based reagent | *Fbxl12 gRNA sequence* | Sequence is from *Tzelepis et al., 2016* | N/A | TTCGCGATGAGCATCTGCA |
| Software, algorithm | Image J | NIH | RRID:SCR_003070 | |
| Software, algorithm | FlowJo | FlowJo | RRID:SCR_008520 | |
| Software, algorithm | Prism | GraphPad | RRID:SCR_002798 | |
| Software, algorithm | Gen5 | Biotek Instruments | RRID:SCR_017317 | |
| Software, algorithm | SeqKit | *Shen et al., 2016* | RRID:SCR_018926 | |
| Software, algorithm | Bowtie | *Langmead et al., 2009* | RRID:SCR_005476 | |
| Software, algorithm | SAMtools | *Li et al., 2009* | RRID:SCR_002105 | |

*Continued on next page*

*Continued*

| Reagent type (species) or resource | Designation | Source or reference | Identifiers | Additional information |
|---|---|---|---|---|
| Software, algorithm | BEDtools | *Quinlan and Hall, 2010* | RRID:SCR_006646 | |
| Other | LSRII Flow cytometer | BD Bioscience | RRID:SCR_002159 | Flow cytometer |
| Other | FACS Celesta Flow Cytometer | BD Bioscience | RRID:SCR_019597 | Flow cytometer |
| Other | FACSAria II Cell Sorter | BD Bioscience | RRID:SCR_018934 | Flow assisted cell sorter |
| Other | Lionheart LX automated microscope | BioTex Instruments | RRID:SCR_019745 | Automated microscope |
| Other | 4-D Amaxa Nucleofecter | Lonza | NA | Nucleofector |

## Cell lines and maintenance

Abelson virus-transformed pre-B cell lines were maintained in DMEM (Thermo Fisher #11960–077) supplemented with 10% fetal bovine serum, 1 X Penicillin-Streptomycin, 2 mM glutamine, 1 mM sodium pyruvate, 1 X nonessential amino acids, and 0.4% beta-mercaptoethanol at 37 °C with 5% $CO_2$. MCF10A cells were maintained in DMEM/F12 (Gibco, #11330032), 5% horse serum, 20 ng/mL EGF, 0.5 μg/mL hydrocortisone, 100 ng/mL cholera toxin, 10 μg/mL insulin, and 1% Penicillin-Streptomycin at 37 °C with 5% $CO_2$. 293T cells were maintained in DMEM (Corning, #10–013 CM) supplemented with 10% fetal bovine serum and 1 X Penicillin-Streptomycin at 37 °C with 5% $CO_2$. MCF10A cell lines were authenticated by STR profiling, and MCF10A and murine cell lines tested negative for mycoplasma contamination.

*Lig4*[-/-] abl pre-B cells contain pCW-Cas9 (addgene, #50661) which expresses cas9 under a doxycycline-induced promoter. To generate single cell clones of *Lig4*[-/-]*:Trp53bp1*[-/-], *Lig4*[-/-]*:Xrcc6*[-/-], *Lig4*[-/-]*:Prkdc*[-/-], and *Lig4*[-/-]*:Fbxl12*[-/-], guide RNAs (gRNAs) against each gene were cloned into pKLV-U6gRNA-EF(BbsI)-PGKpuro2ABFP (addgene, #62348) modified to express human CD2 as a cell surface marker. *Lig4*[-/-] abl pre-B cells were grown in 3 μg/mL of doxycycline for 2 days and then nucleofected with the pKLV-gRNA plasmid using a Lonza Amaxa Nucleofector. The next day, cells were magnetically selected for human CD2 cell surface expression, and selected cells were grown in 3 μg/mL doxycycline overnight. Serial dilution in 96 well plates was used to isolate single cells. After cell growth, potential clones were confirmed to have the gene of interest knocked out by Sanger sequencing or western blotting. *Lig4* deletion was confirmed by PCR as previously described (*Chen et al., 2021a*).

Bulk gene inactivation gRNAs against *Mre11*, *Rbbp8*, and *Xrcc6* were cloned into pKLV-U6gRNA-EF(BbsI)-PGKpuro2ABFP (addgene, #62348). 293T cells were transfected with the pKLV-gRNA plasmid along with lentiviral packaging and lentiviral envelope plasmids. Three days post-transfection, supernatant containing pKLV-gRNA lentivirus was filtered with a 0.45 μm filter. *Lig4*[-/-] cells were resuspended in the filtered viral supernatant supplemented with 5 μg/mL polybrene (Sigma-Aldrich, #S2667) in six-well plates and centrifuged at 1800 RPM for 1.5 hr at room temperature. After spin infection, virally transduced cells were supplemented with DMEM containing 3 μg/mL doxycycline for 3 days before flow cytometry-assisted cell sorting or magnetic-assisted cell sorting based on hCD2 cell surface expression.

## Flow cytometry

Abl pre-B cells were arrested in $G_0$ using 3 μM imatinib (Selleck Chemicals, #S2475) for 48 hr. MCF10A cells were arrested in $G_0$ by withdrawing EGF for 48 hr. To arrest cells in $G_2$, abl pre-B cells were treated with 10 μM RO-3306 (Selleck Chemicals, #S7747) overnight. For experiments analyzing DNA-PKcs and ATM inhibition, 10 μM NU7441 (Selleck Chemicals, #S2638) or 15 μM KU-55933 (Selleck Chemicals, #S1092) was added 1 hr prior to irradiation. After irradiation with 20 Gray, cells were allowed to recover for 3 hr. Cells were then pre-extracted with 0.05% Triton-X 100 (imatinib-treated abl pre-B cells), 0.2% Triton-X 100 (proliferating abl pre-B cells), or 0.5% Triton-X 100 (MCF10A cells) in PBS and fixed with BD Cytofix/Cytoperm solution (BD Biosciences, #554722) containing 4.2% form-aldehyde. Fixed cells were stained with anti-RPA32 (Cell Signaling Technology, #2,208 S) for 2 hr at room temperature, and then treated with a fluorescent conjugated secondary antibody (BioLegend, #405,416 or BioLegend, #405418) for 1 hr at room temperature. 7-AAD was added to each sample

to stain for DNA content. Cells were analyzed using a BD LSRII Flow Cytometer or a BD FACSCelesta and flow cytometry results were further analyzed using FlowJo.

## Nuclear RPA immunofluorescence staining

A total of 60,000 $G_0$-arrested MCF10A cells grown on cover slips were irradiated with 10 Gray IR and then allowed to recover for 3 hr at 37 °C with 5% $CO_2$. Cells were then washed with PBS containing 0.1% Tween-20 (PBST), pre-extracted using cold 0.5% Triton-X100 in PBS for 5 min, fixed with 4% formaldehyde for 15 min, and blocked in 3% BSA-PBST for 1 hr at room temperature. Cells were incubated overnight at 4 °C in primary antibody (anti-RPA32, Cell Signaling Technology, #2208). Samples diluted in 3% BSA-PBST were then washed 3 x with PBST, incubated with secondary antibody diluted in 3% BSA (Alexa Fluor 594 Goat anti-Rat IgG, BioLegend, #405422) in the dark for 1 hr at room temperature, washed 3 x with PBST, and mounted in Prolong Gold Antifade Mountant with DAPI (Life Technologies, #P-36931). Images were taken using a Biotek Lionheart Automatic Microscope and foci quantification was performed using Biotek Gen5 software.

## END-Seq and RPA-ChIP Seq

Sequencing assays were performed in $Lig4^{-/-}$ abl pre-B cells after arrest in $G_0$ with imatinib for 24 hr or arrest in $G_2$ with RO-3306 for 12 hr, then treated with doxycycline (3 µg/µL) for 24 hr followed by tamoxifen treatment (1 µM) for 4 or 8 hr to induce AsiSI breaks in the nucleus. Cell cycle arrest and *AsiSI* induction were confirmed as previously described (*Paiano et al., 2021*). $G_1$ and $G_2$ arrest were confirmed by EdU/DAPI FACS. Cells were pulsed with EdU (10 µM) for 30 min and then fixed in a 1% formaldehyde solution and stained with an AF488 azide. Approximately 90% of cells were in G1 or G2 (respectively) at the time of tamoxifen addition. *AsiSI* induction was confirmed by staining with an anti-phospho-KAP1 antibody (Thermo Fisher A300-767A) at multiple timepoints and then staining with a fluorescent secondary antibody (AF647). *AsiSI* was induced at consistent levels after 4 hr (90–95% positivity) (data not shown). END-Seq was performed as previously described (*Chen et al., 2021a*; *Wong et al., 2021*). Cells were embedded in agarose plugs, lysed, and treated with proteinase K and RNase A. The DNA was then blunted with ExoVII (NEB) and ExoT (NEB), A-tailed, and ligated with a biotinylated hairpin adaptor. DNA was then recovered and sonicated to a length between 150 and 200 bp and biotinylated DNA fragments were purified using streptavidin beads (MyOne C1, Invitrogen). The DNA was then end-repaired and ligated to hairpin adaptor BU2 and amplified by PCR. RPA single-strand DNA sequencing was performed as previously described (*Paiano et al., 2021*). Cells were fixed in 1% formaldehyde (Sigma F1635) for 10 min at 37 °C, quenched with 125 mM glycine (Sigma), washed twice with cold 1×PBS. After centrifugation, pellets were frozen on dry ice, and stored at −80 °C. Sonication, immunoprecipitation, and library preparation were performed as previously detailed (*Tubbs et al., 2018*). Before immunoprecipitation, sheared chromatin was precleared with 40 µL of Dynabeads Protein A (Thermo Fisher) for 30 min at 4 °C. Sheared chromatin was enriched with 10 µg of anti-RPA32/RPA2 antibody (Abcam ab10359) on Dynabeads Protein A overnight at 4 °C. During library preparation, kinetic enrichment of single-strand DNA was performed by heating sheared DNA for 3 min at 95 °C and allowing DNA to return to room temperature (*Tubbs et al., 2018*). All END-seq and RPA ChIP-seq libraries were collected by gel purification and quantified using qPCR. Sequencing was performed on the Illumina NextSeq500 (75 cycles) as previously described (*Chen et al., 2021a*).

## Genome alignment and visualization

END-seq and RPA ChIP-seq single-end reads were aligned to the mouse genome (mm10) using Bowtie v1.1.2 (*Langmead et al., 2009*) with parameters (-n 3 k 1 l 50) for END-seq and (-n 2 m 1 l 50) for RPA ChIP-seq. All plots or analysis were done for the top 200 *AsiSI* sites determined by END-seq. Alignment files were generated and sorted using SAMtools (*Li et al., 2009*) and converted to bedgraph files using bedtools genomecov *Quinlan and Hall, 2010* following by bedGraphToBigWig to make a bigwig file (*Kent et al., 2010*). Visualization of genomic profiles was done by the UCSC genome browser (*Kent et al., 2002*) and normalized to present RPM. Heat maps were produced using the R package pheatmap.

## Genome-wide guide RNA library screen

A total of 144 million $Lig4^{-/-}$ abl pre-B cells with tet-inducible Cas9 were transduced with a lentiviral gRNA library (Pooled Library #67988, Addgene) containing 90,000 gRNAs targeting over

18,000 mouse genes. Three days post-infection, cells were sorted for gRNA vector expression using a BD FACSAria flow cytometry assisted cell sorter by BFP fluorescence. The next day, sorted cells were treated with 3 μg/ml doxycycline to induce Cas9-mediated gene inactivation. Seven days later, cells were treated with imatinib to arrest cells in $G_0$. Forty-eight hours later, cells were irradiated with 20 Gray and allowed to recover for 2 hr. After collection, cells were permeabilized, fixed, and stained with anti-RPA32 in the same manner as described in the Flow Cytometry section. After staining, the top 10% and bottom 10% of RPA stained cells were collected using flow cytometry assisted cell sorting and genomic DNA was extracted. An Illumina sequencing library was generated using two rounds of PCR to amplify the gRNA and add a barcode, then purified PCR products containing the barcoded enriched gRNAs were sequenced on an Illumina HiSeq2500. Sequencing data were processed as previously described (*Chen et al., 2021a*).

## Western blotting

The following antibodies were used for western blot analysis: CtIP (gift from Dr. Richard Baer, [Columbia University, New York], 1:1000), MRE11 (Novus Biologicals, NB100-142, 1:2000), GAPDH (Sigma, G8795, 1:10,000), DNA-PKcs (Invitrogen, MA5-32192, 1:1000), KAP1 (Genetex, GTX102226, 1:2000), KU70 (Cell Signaling Technology, #4588, 1:1000).

Plasmid Constructs pCW-Cas9 was a gift from Eric Lander and David Sabatini (Addgene plasmid #50661) (*Wang et al., 2014*). pKLV-U6gRNA(BbsI)-PGKpuro2ABFP was a gift from Kosuke Yusa (Addgene plasmid #50946) (*Koike-Yusa et al., 2014*). Mouse Improved Genome-wide Knockout CRISPR Library v2 was a gift from Kosuke Yusa (Addgene #67988) (*Tzelepis et al., 2016*).

## Acknowledgements

The authors thank Chitra Mohan for designing the model graphic and Yinan Wang for performing the bioinformatics for the high throughput screen. We thank the Weill Cornell Flow Cytometry Core for flow cytometric analysis and the Weill Cornell Epigenomics Core for providing advice and performing the sequencing for the high throughput screen. JKT is supported by NIH R35 GM139816 and R01 CA95641. BPS is supported by NIH R01 AI047829 and R01 AI074953. FCF is supported by NIH F31 CA239442. JKT and BPS were also supported by the Starr Cancer Consortium and Emerson Collective Cancer Research Fund.

## Additional information

### Competing interests

Jessica K Tyler: Senior editor, *eLife*. The other authors declare that no competing interests exist.

### Funding

| Funder | Grant reference number | Author |
|---|---|---|
| NIH Office of the Director | R35 GM139816 | Jessica K Tyler |
| NIH Office of the Director | RO1 CA95641 | Jessica K Tyler |

The funders had no role in study design, data collection and interpretation, or the decision to submit the work for publication.

### Author contributions

Faith C Fowler, Data curation, Formal analysis, Investigation, Methodology, Writing - original draft; Bo-Ruei Chen, Conceptualization, Writing - review and editing; Nicholas Zolnerowich, Data curation, Investigation; Wei Wu, Raphael Pavani, Jacob Paiano, Chelsea Peart, Zulong Chen, Investigation; André Nussenzweig, Funding acquisition, Project administration; Barry P Sleckman, Conceptualization, Project administration; Jessica K Tyler, Conceptualization, Funding acquisition, Writing - review and editing

### Author ORCIDs

Faith C Fowler (iD) http://orcid.org/0000-0002-7180-8141
Bo-Ruei Chen (iD) http://orcid.org/0000-0001-6404-2099

Barry P Sleckman [iD]http://orcid.org/0000-0001-8295-4462
Jessica K Tyler [iD]http://orcid.org/0000-0001-9765-1659

**Decision letter and Author response**
Decision letter https://doi.org/10.7554/eLife.74700.sa1
Author response https://doi.org/10.7554/eLife.74700.sa2

## Additional files

### Supplementary files
• Transparent reporting form

### Data availability
Sequencing data have been deposited in GEO under accession codes GSE186087.

The following dataset was generated:

| Author(s) | Year | Dataset title | Dataset URL | Database and Identifier |
|---|---|---|---|---|
| Wu Wei, Fowler FC, Bo-Ruei Chen, Zolnerowich N, Pavani R, Paiano J, Peart C, Chen Z, Nussenzweig A, Tyler JK, Sleckman BP | 2022 | DNA-PK Promotes DNA End Resection at DNA Double Strand Breaks in G0 cells | https://www.ncbi.nlm.nih.gov/geo/query/acc.cgi?acc=GSE186087 | NCBI Gene Expression Omnibus, GSE186087 |

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
