## [Editor Report]

This manuscript will be of relevance to scientists interested in cell cycle, DNA repair, and genome stability reporting the unexpected discovery that the DNA-dependent protein kinase (DNA-PK) is required for DSB resection in G0 cells, whereas it is known and confirmed here that it inhibits resection in G1 and G2 cells. This finding has important implications for the clinical application of DNA-PK-targeted inhibitors. The data are of high quality and derive from two independent cell lines, genetic requirements were mostly established by gene knockouts, and the latest genome-wide sequencing techniques were applied to measure resection tracts. The key claims of the manuscript are supported by the data presented by the authors.

---

## [Decision Letter]

**Decision letter after peer review:**

Thank you for submitting your article "DNA-PK Promotes DNA End Resection at DNA Double Strand Breaks in G_0_ cells" for consideration by *eLife*. Your article has been reviewed by 3 peer reviewers, including Wolf-Dietrich Heyer as the Reviewing Editor and Reviewer #1, and the evaluation has been overseen by Kevin Struhl as the Senior Editor.

Recommendations to the authors

All reviewers agree that the authors made a significant discovery that will be of great interest to the field. There was also a consensus that more data are needed in three areas. (1) More evidence is needed to ensure that the observations apply to human cells, in light of the fact that human cells have 50x less DNA-PK than murine cells. (2) More evidence is needed that the phenomenon is truly G0 specific and not occurring in G1 cells. (3) There is some concern that most experiments were conducted in LIG4-deficient cells. Without wanting to be prescriptive, additional data with the LIG4-proficient human MCF10A cell line applying END-Seq to G0 and G1 cells evaluating DSB resection in dependence of DNA PK would nicely address these concerns.

Essential revisions

1) Despite showing a similar increase in the chromatin-bound RPA upon irradiation in WT abl pre-B cells, the authors utilize for the majority of the experiments abl pre-B cells deficient for Lig4. It is important to verify that the inability to complete the NHEJ repair process due to the lack of Lig4 does not cause pathological resection of the DSBs. Additional examination in NHEJ proficient cells would eliminate this concern.

2) The authors use Imatinib treatment for abl pre-B cells or serum deprivation in MCF10A for 48 h to arrest cells in G0 phase. However, further analyses are needed to validate that after only two days of treatment cells have really exited the cell cycle.

3) The authors suggest that the DNA-PK complex promotes DSBs resection uniquely in G0 but not in other cell cycle phases. To draw this conclusion, a few control experiments should be included. END-seq analysis of AsiSI-induced DSBs in Lig4-/- abl pre-B cells should be performed also in G1 phase. Moreover, taking into account that V(D)J recombination is happening in the G0/G1 phase of immune cells, which could explain the high level of resection shown in G1 pre-B cells, the experiments should be replicated in another cell line, like in MCF10A.

4) The paper nicely demonstrates the involvement of the DNA-PK complex in DSBs resection in G0 murine abl pre-B cells. To consolidate the results, the authors should confirm the requirement of the DNA-PK complex (including KU70/80) and its regulation by FBXL12 also in MCF10A. This extends their finding from a murine model to human cells and is important to expand the impact of the paper.

5) The authors should extract more data from the END-seq analyses and report average and median resection tracts for each experiment.

6) In the discussion about the mechanistic significance for G0 DSB resection, the authors may want to consider adding a paragraph on TMEJ and the possibility that end-processing represents a way to shift an NHEJ substrate to a TMEJ substrate. The authors may also want to consider and discuss a relatively recent report on resection dependent classic NHEJ (https://pubmed.ncbi.nlm.nih.gov/28132842/).

7) CtIP is licensed by phosphorylation in S/G2. Can the authors comment on CtIP activity in G0/G1 and the possibility that some phosphorylation exists under these conditions?

8) Chromatin-bound RPA levels have been assessed in various experiments throughout the manuscript. However, the dose of X-rays and the time after irradiation have been frequently changing. Please clarify the reason for those inconsistencies.

9) A genome wide CRISPR/Cas9 screen was performed to identify factors promoting resection in G0 cells. Detailed information regarding the screen is however missing. Please specify how many times it was repeated and clarify how the fold enrichment of the gRNAs is calculated.

10) A clearer description of the Lig4-/- AsiSI abl pre-B cells is needed. Please specify how these cells were generated and how DSB induction was optimized. Please add also information on the concentration of tamoxifen utilized. Furthermore, the location of the DSBs should be analyzed to show if there is a subgroup of them which is prone to resection and is perhaps located in a particular genomic region (genes versus non-genes, transcriptionally active versus inactive, etc.).

11) In the discussion, the authors mention Artemis as an example of a nuclease involved in resection and V(D)J recombination. Since Artemis is known to be involved in NHEJ in G1, and in HR in G2, its function could be tested to assess the difference in NHEJ repair in quiescent and G1 cells and would serve as a nice control to show the processes are entirely different.

[Editors' note: further revisions were suggested prior to acceptance, as described below.]

Thank you for resubmitting your work entitled "DNA-PK Promotes DNA End Resection at DNA Double Strand Breaks in G_0_ cells" for further consideration by *eLife*. Your revised article has been evaluated by Kevin Struhl (Senior Editor) and a Reviewing Editor.

The manuscript has been improved but there are some remaining issues that need to be addressed, as outlined below:

The revised manuscript strengthened an already strong manuscript, and the revision addresses the concerns of the reviewers. There is one point of clarification needed about Figure 4E. The rebuttal and text describe that G0 wild type MCF10A cells show DNA PK dependent resection. It follows that upon DNAPK inhibition the amount of RPA should decrease, as resection is reduced. As far as I understand Figure 4E, the results show the opposite: the green curve (DNAPKi +IR) shifts to the right (more RPA) compared to IR only. IS there an error in the figure, in the description, or in my understanding?

---

## [Author Response]

Essential revisions1) Despite showing a similar increase in the chromatin-bound RPA upon irradiation in WT abl pre-B cells, the authors utilize for the majority of the experiments abl pre-B cells deficient for Lig4. It is important to verify that the inability to complete the NHEJ repair process due to the lack of Lig4 does not cause pathological resection of the DSBs. Additional examination in NHEJ proficient cells would eliminate this concern.

In addition to showing that chromatin-bound RPA increases upon irradiation in quiescent wild type (WT) abl pre-B cells (Figure 1 —figure supplement 1B) we now show that DNA-PK activity promotes DNA end resection in quiescent WT abl pre-B cells. This is included as new Figure 2 —figure supplement 1C. In addition to showing that chromatin-bound RPA increases upon irradiation in quiescent WT human MCF10A Figure 1 —figure supplement 1E, we show that RPA foci formation increases after IR in quiescent WT MCF10A (Figure 1C), the dependance of DNA-PK activity for RPA foci formation in quiescent WT MCF10A (Figure 2 —figure supplement 1D) and that DNA-PK promotes chromatin bound RPA in quiescent WT MCF10A cells, but not G_1_ WT RPA foci formation increases after IR in WT MCF10A (new figure 4E).

2) The authors use Imatinib treatment for abl pre-B cells or serum deprivation in MCF10A for 48 h to arrest cells in G0 phase. However, further analyses are needed to validate that after only two days of treatment cells have really exited the cell cycle.

This is a good point. Accordingly, we recently published an extensive analysis of phospho targets that demonstrates that Imatinib treatment in abl pre-B cells and serum deprivation in MCF10A for 48 h arrests cells in G0 phase not G1 phase (Chen et al. 2021 *eLife*), using the identical conditions for arrest that we use in the current manuscript.

3) The authors suggest that the DNA-PK complex promotes DSBs resection uniquely in G0 but not in other cell cycle phases. To draw this conclusion, a few control experiments should be included. END-seq analysis of AsiSI-induced DSBs in Lig4-/- abl pre-B cells should be performed also in G1 phase. Moreover, taking into account that V(D)J recombination is happening in the G0/G1 phase of immune cells, which could explain the high level of resection shown in G1 pre-B cells, the experiments should be replicated in another cell line, like in MCF10A.

We had shown that chromatin-bound RPA increases after IR in G_1_ phase cells by gating the cycling cells on 2N DNA content at the time of the analysis. RPA only binds to ssDNA, so there is no other conclusion to be made other than this is due to increased DNA end resection. It is not possible for us to perform the reviewers suggested END-seq in G_1_ phase cells, because we sort for cycling G_1_ cells, and by the time that the END-Seq analysis would be done the cells would no longer be in G_1_ phase, given that the induction of *AsiSI* is by treatment with doxycycline for 24 hours followed by nuclear localization of *AsiSI* by tamoxifen treatment for 4 and 8 hours to induce *AsiSI* cutting. Regarding the question of whether the RPA binding is due to V(D)J recombination, we have published extensively that VDJ occurs in the imatinib (G_0_) arrested cells while the RPA binding to chromatin and end-resection that we see in G_0_ arrested cells is totally dependent on irradiation. Nonetheless, we now show that chromatin-bound RPA increases in both G_0_ and G_1_ WT MCF10A cells after IR but is only dependent on DNA-PK in G_0_ cells (new Figure 4E).

4) The paper nicely demonstrates the involvement of the DNA-PK complex in DSBs resection in G0 murine abl pre-B cells. To consolidate the results, the authors should confirm the requirement of the DNA-PK complex (including KU70/80) and its regulation by FBXL12 also in MCF10A. This extends their finding from a murine model to human cells and is important to expand the impact of the paper.

Unfortunately, human cells are more sensitive to DNA-PK depletion than murine cells, so genetic knockout of DNA-PK was unsuccessful using our lab’s CRISPR/Cas9 technique. Previous literature shows that Ku70 and Ku80 are encoded by essential genes and a homozygous knockout is difficult to achieve (Hendrickson, E.A., et al. 2008 PNAS; Hendrickson, E.A. et al. 2002 PNAS). However, as discussed above, we now include new data showing DNAPK dependent resection in MCF10A cells in G_0_ phase (Figure 4E). We attempted to make MCF10A *Fbxl12^-/-^* cells numerous times with different gRNAs, and screened dozens of clones, but were unsuccessful at obtaining homozygous knockout clones. This is what has held up resubmission of our paper for so long. We are still attempting to make homozygous knockout clones but would add at a minimum 2 more months before our resubmission.

5) The authors should extract more data from the END-seq analyses and report average and median resection tracts for each experiment.

This has been done and included as Figure 4D.

6) In the discussion about the mechanistic significance for G0 DSB resection, the authors may want to consider adding a paragraph on TMEJ and the possibility that end-processing represents a way to shift an NHEJ substrate to a TMEJ substrate. The authors may also want to consider and discuss a relatively recent report on resection dependent classic NHEJ (https://pubmed.ncbi.nlm.nih.gov/28132842/).

We now include some discussion about these possibilities. TMEJ requires some DNA end resection and is considered a backup pathway for breaks that cannot complete HR but have undergone DNA end resection or at breaks that do not have suitable ends for NHEJ. It is possible, especially in *Lig4^-^*^/-^ cells, that TMEJ could be responsible for DSB repair after resection in G_0_, though we note that TMEJ is Ku-independent and our G_0_ DNA end resection is Ku-dependent.

We now mention resection-dependent NHEJ as an example of nuclease activity in DSB repair outside of HR, but the reviewer cited paper shows that Artemis is absolutely required for this process, and our data shows Artemis has no role in G_0_ DSB repair (Figure 1—figure supplement 1D), it is unlikely this repair process is what we observe in G_0_ cells.

7) CtIP is licensed by phosphorylation in S/G2. Can the authors comment on CtIP activity in G0/G1 and the possibility that some phosphorylation exists under these conditions?

We now mention in the introduction that CtIP has been shown to be phosphorylated by PLK3 in G_1_ during DSB repair and have discussed the likelihood that CtIP is also phosphorylated in G_0_ cells.

8) Chromatin-bound RPA levels have been assessed in various experiments throughout the manuscript. However, the dose of X-rays and the time after irradiation have been frequently changing. Please clarify the reason for those inconsistencies.

Most flow cytometry experiments were performed with 20 Gray IR and samples were collected 3 hours after irradiation. Figures 1B, 2C, 3D, 4E exhibit either a slightly different dose of IR (15 Gy) or time point (2 hours or 4 hours) due to these experiments being performed at a different facility. The experimental parameters were reassessed and optimized at the new facility. The CRISPR/Cas9 screen was collected at 2 hours instead of 3 due to time constraints of the experiment. We have observed that the extent of resection in G_0_ is not significantly different between 2-4 hours after IR. For immunofluorescence experiments doses of IR were experimentally determined to provide resolution of the RPA foci by IF to enable counting, which required a lower IR dose. Immunofluorescence after 15-20 Gray showed RPA foci merging to fill the entire nucleus.

9) A genome wide CRISPR/Cas9 screen was performed to identify factors promoting resection in G0 cells. Detailed information regarding the screen is however missing. Please specify how many times it was repeated and clarify how the fold enrichment of the gRNAs is calculated.

We now include in the figure legend how the fold enrichment was calculated: the ratio of normalized read number of gRNAs in the low RPA population and that in the high RPA population and vice versa. We also indicate that the screen was performed once (n=1). Notably, we validated all the hits that we studied in this manuscript.

10) A clearer description of the Lig4-/- AsiSI abl pre-B cells is needed. Please specify how these cells were generated and how DSB induction was optimized. Please add also information on the concentration of tamoxifen utilized.

We now include this information in the methods section.

Furthermore, the location of the DSBs should be analyzed to show if there is a subgroup of them which is prone to resection and is perhaps located in a particular genomic region (genes versus non-genes, transcriptionally active versus inactive, etc.).

Generally, there are about 200 broken *AsisI* breaks detectable by ENDseq, so we used the top 200 sites. Most of them are close to TSS: 189 out of 200 are within +/2kb of TSS. And 170 of the 200 AsisI sites that their closest genes are expressed (RPKM>0.1) measured by a nasRNA-seq in abl pre-B cells. These analyses are now included in Figure 1 – supplemental figure 2.

11) In the discussion, the authors mention Artemis as an example of a nuclease involved in resection and V(D)J recombination. Since Artemis is known to be involved in NHEJ in G1, and in HR in G2, its function could be tested to assess the difference in NHEJ repair in quiescent and G1 cells and would serve as a nice control to show the processes are entirely different.

To address this comment, we have now tested Artemis dependency in G_0_ arrested abl pre-B cells after IR and show that it is not required for resection in G_0_ arrested cells in figure 1 —figure supplement 1D. We have altered the discussion accordingly.

[Editors' note: further revisions were suggested prior to acceptance, as described below.]

The revised manuscript strengthened an already strong manuscript, and the revision addresses the concerns of the reviewers. There is one point of clarification needed about Figure 4E. The rebuttal and text describe that G0 wild type MCF10A cells show DNA PK dependent resection. It follows that upon DNAPK inhibition the amount of RPA should decrease, as resection is reduced. As far as I understand Figure 4E, the results show the opposite: the green curve (DNAPKi +IR) shifts to the right (more RPA) compared to IR only. IS there an error in the figure, in the description, or in my understanding?

Thank you for carefully reading our manuscript and looking at the new data. It transpires that somehow upon Figure assembly we did invert the labelling somehow. Thanks so much for seeing this as it avoids an embarrassing correction. We have updated the figure accordingly.